# Hydrogel Formation of Enzymatically Solubilized Corn Bran Feruloylated Arabinoxylan by Laccase-Catalyzed Cross-Linking

**DOI:** 10.3390/foods14162819

**Published:** 2025-08-14

**Authors:** Changxin Liu, Zifan Zhao, Weijie Zhong, Zilong Su, Qing Zhang, Yiqing Zhang, Shang Lin, Xuesong Lu, Wen Qin

**Affiliations:** 1College of Food Science, Sichuan Agricultural University, Ya’an 625014, China; z2037403620@163.com (C.L.); 2024218014@stu.sicau.edu.cn (Z.Z.); 17781616764@163.com (W.Z.); 18862082773@163.com (Z.S.); zhangqing@sicau.edu.cn (Q.Z.); scyszyq@163.com (Y.Z.); shanli@sicau.edu.cn (S.L.); 2College of Culinary and Food Science Engineering, Sichuan Tourism University, Chengdu 610100, China

**Keywords:** arabinoxylan, ferulic acid, diferulic acid, hydrogel, laccase

## Abstract

In order to upgrade the potential of cereal bran arabinoxylan for advanced hydrogel applications, a deep understanding of its gelation process is required. This work provides a comprehensive and systematic analysis of the laccase-catalyzed cross-linking of feruloylated arabinoxylan (FAX) to establish a clear link between processing conditions and final hydrogel properties. Endo-1,4-xylanase was used to obtain corn bran FAX rich in ferulic acid moieties, and then we demonstrated that gel formation is driven by the oxidative coupling of these feruloyl monomers into diferulic acid bridges, e.g., 8-5′, 5-5′, 8-*O*-4′, and 8-5′ benzofuran diferulic acids. A systematic investigation revealed that hydrogel properties were significantly affected by the processing conditions, i.e., FAX concentration, enzyme dosage, reaction pH, and reaction temperature during the enzymatic gel formation catalyzed by laccase. While gel strength peaked at a FAX concentration of 30 mg/mL, an optimal temperature of 25 °C and pH 6 were identified. Notably, we discovered a critical trade-off with enzyme concentration: higher laccase levels accelerated the reaction but compromised the final hydrogel’s mechanical strength and water retention. Gelation failed completely at pH ≥ 9 due to laccase inactivation. Meanwhile, scanning electron microscope analysis revealed that the microstructure of the FAX hydrogels was significantly affected by changes in the processing conditions. These findings offer crucial insights for the rational design of FAX-based hydrogels, enabling their tailored fabrication for food industry applications.

## 1. Introduction

Hydrogels are three-dimensional polymeric networks capable of absorbing and retaining significant volumes of water or biological fluids without dissolving. Due to their unique properties, they are widely utilized in diverse applications across the pharmaceutical and food industries, e.g., drug delivery [1,2], bone defect repair [3], wound healing [4], biosensors [5], traditional hydrogel-like food, e.g., jelly [6], and as thickening agents, gelling agents, and food stabilizers [6]. Among the various types, polysaccharide-based hydrogels have garnered considerable research attention owing to their inherent features of biodegradability, high hydrophilicity, and excellent biocompatibility [6,7,8]. Furthermore, natural polysaccharides exhibit great nutritional functions, e.g., antioxidant activity and immunomodulatory effects [9,10,11,12].

Numerous natural polysaccharides, such as alginate, chitosan, hyaluronic acid, cellulose, pectin, glucogalactan, and konjac glucomannan, have been extensively studied for their ability to form hydrogels through diverse mechanisms [6,7,8,13]. Cereal bran, a major by-product of the agro-industry, represents a huge source of such underutilized polysaccharides. It is predominantly composed of dietary fiber, approximately 40–60% of its total mass, of which xylan is the most abundant fraction [14,15,16]. As the second most abundant polysaccharide in nature after cellulose, xylan constitutes 20–40% of the bran by weight, depending on the botanical source [14,15,16]. Hence, developing hydrogels from this abundant biopolymer has become an area of increasing interest.

The fundamental structure of arabinoxylan, the primary form of xylan in cereal brans, comprises a β-1,4-linked xylopyranosyl backbone decorated with arabinofuranosyl side chains at the C-2 and/or C-3 positions. The degree of substitution is expressed as the arabinose-to-xylose (A/X) ratio, typically ranging from 0.5 to 1.2 [16,17]. A unique feature of arabinoxylan is the presence of feruloyl groups, which are ester-linked to the C-5 position of some arabinose units, providing the polysaccharide with biological activities such as antioxidant and immunomodulatory effects [17]. These feruloyl groups are particularly crucial for gelation, as they can undergo oxidative coupling to form diferulic acid bridges such as 8-5′, 5-5′, and 8-*O*-4′ diferulic acids [18,19]. This reaction serves as a covalent cross-link between two adjacent arabinoxylan chains, thereby forming a stable three-dimensional network referred to as a “sandwich structure” [17]. While hydrogel formation from xylan can be achieved through various means, including chemical crosslinking [20,21], physical interactions [22], or synergistic effects with other macromolecules such as protein and starch [23,24,25,26,27], the cross-linking mediated by feruloyl moieties is of particular interest. This specific gelation can be initiated either enzymatically, e.g., using laccase or peroxidase or through chemical oxidation [17]. Among the available methods for arabinoxylan gelation, the enzymatic approach, particularly using laccase, is increasingly recognized for its distinct advantages [19,28,29,30,31,32,33,34]. This method is favored because it proceeds under mild conditions with high specificity, aligns with green chemistry principles by generating minimal by-products, and yields hydrogels with excellent biocompatibility, high food-grade safety, and a notably stable network structure, thus attracting significant research attention [19,28,29,30,31,32,33,34].

The gelation behavior of arabinoxylan is dependent on its structural features, including molecular weight and A/X ratio [19,34,35]. For enzymatically catalyzed hydrogel formation, however, the degree of feruloylation is the most key factor [36]. Moreover, the content of these crucial groups is highly dependent on the extraction method used. For instance, conventional chemical or physical extraction methods often cleave the ester bonds that link ferulic acid to the arabinoxylan backbone, leading to a substantial loss of feruloylation. This reduced degree of feruloylation impairs subsequent hydrogel formation, stability, and strength. In contrast, employing a milder enzymatic approach, specifically using 1,4-β-xylanases, offers a more effective strategy for preserving these essential feruloyl moieties. Laccase is an oxidoreductase widely employed for phenolic compound oxidation [19,32,33]. Despite its capacity to crosslink ferulic acid, as has been reported, a systematic study for developing laccase-catalyzed arabinoxylan hydrogel formation is still required. Specifically, there is a lack of clear understanding of how key processing parameters, including substrate concentration, enzyme dosage, temperature, and pH, synergistically affect the microstructure, mechanical strength, and water-holding capacity of the arabinoxylan hydrogels.

Therefore, this study aimed to systematically investigate the mechanism of laccase-mediated hydrogel formation from arabinoxylan. To this end, arabinoxylan with a desirable degree of feruloylation was first prepared via a targeted enzymatic extraction using 1,4-β-xylanase. Subsequently, we investigated the effect of four key processing parameters, i.e., arabinoxylan concentration, laccase concentration, reaction temperature, and pH, on the physical properties of arabinoxylan hydrogels. The rheological properties, hydrogel strength, and microstructure of the hydrogels were characterized using a rheometer, a texture analyzer, and a scanning electron microscope (SEM), respectively. The objective of this study was to elucidate the relationships between processing parameters and the gelling properties of feruloylated arabinoxylan hydrogels, and to advance the understanding of the controllable fabrication of functional arabinoxylan hydrogels for both the food and pharmaceutical industries.

## 2. Materials and Methods

### 2.1. Materials and Chemicals

Corn bran was purchased from Guizhou Qiannan Qianshun County Jinyou Livestock Co. (Qiannan, China). The corn bran sample was soaked with 95% ethanol to remove the polyphenols and reducing sugars and was further de-starched by α-amylase (15 U/mL) (Beijing Solarbio Science & Technology Co., Ltd., Beijing, China). Protein was removed by incubation with pancreatin protease (5 U/mL) at 40 °C for 24 h. The enzymes were inactivated by heating at 100 °C for 30 min, and the resulting pellet was freeze-dried. This destarched corn bran sample was denoted as DCB. Endo-1,4-β-xylanase from *Thermomyces lanuginosus* (*Tl*Xyn11) and laccase from *Trametes versicolor* were purchased from Merck (Darmstadt, Germany). All other chemicals were purchased from Sigma-Aldrich (Burlington, MA, USA).

### 2.2. Enzymatic Preparation of Soluble Feruloylated Arabinoxylan

The soluble feruloylated arabinoxylan was enzymatically extracted from 10% (*v*/*v*) DCB substrate by *Tl*Xyn11 (2.5 U/mL) in sodium acetate buffer (50 mM, pH 6.0) at 50 °C for 24 h. After centrifugation at 8000 rpm for 30 min, the supernatant was collected and precipitated with 3 volumes of 95% ethanol at 4 °C overnight. The precipitate was dissolved in Milli-Q water and then dialyzed for 2 days using a 3.5 kDa cut-off dialysis bag. Afterwards, the samples were freeze-dried and stored at −20 °C for further analysis and experiments. This sample was referred to as FAX.

### 2.3. Carbohydrate Analysis of Soluble Feruloylated Arabinoxylan

The total polysaccharide content and total protein content were determined according to a previous study [12]. The monosaccharide composition was determined by a previous study using high-performance liquid chromatography equipped with a diode array detector (HPLC-DAD, Agilent 1260, Agilent Technologies, Palo Alto, CA, USA) [12]. The polysaccharide sample was hydrolyzed by trifluoroacetic acid and then derivatized with 0.5 M phenyl-3-methyl-5-pyrazolone (PMP). The sample was eluted with 83% PBS (0.1 M, pH 6.7) and 17% acetonitrile through a column (InfinityLab Poroshell 120 EC-C18, 4 µm, 4.6 × 150 mm, Agilent Technologies, Inc., Palo Alto, CA, USA) at a flow rate of 1.0 mL/min at 30 °C, and analyzed by the DAD system at a wavelength at 245 nm. The molecular weight of FAX was determined by high-performance size exclusion chromatography coupled with multi-angle laser light scattering and a refractive index detector (HPSEC-MALLS-RID, Wyatt Technology Co., Santa Barbara, CA, USA) according to a reported study [12].

### 2.4. Analysis of Ferulic Acid and Diferulic Acid of FAX Samples

The FAX sample (final concentration of 10 mg/mL) was saponified gently with NaOH (final concentration of 1 M) at 4 °C for 10 h. The reaction was stopped by addition of 6 M HCl for neutralization. The mixture was filtered through a 0.22 µm filter to collect the free ferulic acid and diferulic acid. The ferulic acid and diferulic acid contents of FAX samples were analyzed and quantified by a quadrupole time-of-flight mass spectrometer (Q-ToF-MS, Agilent G6545, Palo Alto, CA, USA) according to a previous study with some modifications [37]. In brief, 10 µL of sample was injected onto the Agilent ZORBAX Eclipse Plus C18 column (2.1 × 50 mm, Agilent G6545, Palo Alto, CA, USA). Elution was achieved at a flow rate of 0.3 mL/min and 40 °C column temperature using acetonitrile (eluent A) and 0.1% (*v*/*v*) aqueous formic acid (eluent B) with the following gradient program: 0–4.5 min, 10% A/90% B; 4.5–14.5 min, gradient up to 46% A/54% B; 14.5–15 min, gradient up to 100% A/0% B and continue for 2 min until 17.0 min; 17–17.1 min, gradient to 10% A/90% B and continue for 3 min until a total runtime of 20.1 min. The electrospray was operated in negative scan mode with the following settings: capillary voltage, 4.5 kV; end plate off set, 0.5 kV; nebulizer pressure, 3.0 bar; dry gas flow, 12.0 L/min; and dry gas temperature, 280 °C.

### 2.5. Hydrogel Formation of Feruloylated Arabinoxylan

The FAX solution was prepared by dissolving the sample in 50 mM sodium acetate buffer (pH 6.0) to a final concentration of 40 mg/mL, followed by equilibration to 25 °C for 30 min. Laccase from *Trametes versicolor* was then added to a final concentration of 1.25 mg/mL, and the mixture was incubated at 25 °C for 1 h with continuous shaking. Samples were collected after 2, 5, 10, 30, and 60 min and immediately frozen to stop the reaction. To examine the effects of key parameters on hydrogel formation, arabinoxylan concentration (10–50 mg/mL), laccase concentration (0.625–10 mg/mL), pH (3–10), and temperature (15–55 °C) were systematically varied in separate experiments.

### 2.6. Rheological Properties Analysis

The rheological behaviors of FAX hydrogel samples were analyzed using a modular compact rheometer (MCR 102e, Anton Paar GmbH, Graz, Austria) equipped with a parallel plate at 25 °C (50 mm diameter) with a probe selection of PP20 and a gap of 1 mm according to a reported study [12]. After temperature equilibration at 25 °C for 2 min, time sweep measurements (0–60 min) were performed at 1 Hz and 0.1% strain to monitor the evolution of storage (G′) and loss (G″) moduli. Subsequently, strain amplitude sweeps (0.01–100%) were conducted at 1 Hz to record both moduli as functions of applied strain. Finally, frequency sweeps (0.1–10 Hz) were performed at 0.1% constant strain to determine the frequency dependence of G′ and G″.

### 2.7. Water Holding Capacity

The water holding capacity (WHC) of each FAX hydrogel sample was determined by a previously reported method [12]. WHC was calculated using the following equation:$$\mathrm{WHC}\;\left(\%\right)\;=\;\frac{W_{2}\;–{\;W}_{0}}{W_{1}\;–\;W_{0}}\;\times \;100\%$$
where W_0_ is the weight of the centrifuge tube without the hydrogel sample, and W_1_ and W_2_ are the weights of the centrifuge tube with the hydrogel sample before centrifugation, and after centrifugation.

### 2.8. SEM Analysis

Scanning electron microscopy (SEM) was performed using a ZEISS Merlin tabletop microscope (Jena, Germany) [12]. Freeze-dried hydrogel samples were sputter-coated with a thin gold–palladium film, and images were acquired at an accelerating voltage of 3.0 kV to examine their microstructure.

### 2.9. Structural Characterisation of Hydrogels

The FAX hydrogel formed in optimal conditions—reaction time 1 h, substrate concentration 40 mg/mL, laccase concentration 1.25 mg/mL, reaction pH 6, and reaction temperature 25 °C—was analyzed by infrared spectroscopy (FTIR), X diffraction (XRD) and low-field NMR (LF-NMR).

Infrared spectroscopy (FTIR) analysis: the FTIR spectra of each sample was determined by Fourier transform infrared spectroscopy, and prior to the determination, the hydrogels were lyophilized and then ground into powder. The infrared spectra of the hydrogel were tested by adding a KBr press at 1:100 (m:m) with a scanning range of 400–4000 cm^−1^ and a resolution of 4 cm^−1^.

X diffraction (XRD) analysis: The crystal structure of the dried hydrogel samples was determined using an X-ray diffractometer (Dandong Tongda TD-3700, Liaoning, China) under radiation with an operating current of 40 mA and a voltage of 40 kV. The scanning speed was 2°/min and the scanning scattering angle ranged from 5–80°.

Low-field NMR (LF-NMR) analysis: The distribution of free and bound water in the hydrogel samples was determined using a multidimensional microimaging analyzer (NMI20-040V-I, Suzhou, China). The parameters were set as follows: 250 kHz for SW, 1 ms for RFD, 9.00 μs for P1, 14.98 μs for P2, 7000 ms for TW, 0.200 ms for TE, 16,000 for NECH, and 4 scans.

### 2.10. Statistical Analysis

All experiments were performed in triplicate. Data were analyzed by one-way analysis of variance (ANOVA) using SPSS 26.0 (IBM, New York, NY, USA). Mean comparisons were performed with Duncan’s multiple range test, and differences were considered significant at *p* < 0.05.

## 3. Results and Discussions

### 3.1. Structural Characteristics of Feruloylated Arabinoxylan

Soluble feruloylated arabinoxylan was enzymatically extracted from de-starched corn bran using 1,4-β-xylanase, with a yield of 0.75% (Table 1). The total polysaccharide and protein contents were determined to be 80.1% and 2.3%, respectively, indicating that the extracted FAX had high purity (Table 1). As shown in Figure 1a, the SEC chromatogram presented a single peak with a *M*_w_ of 339.8 kDa, indicating FAX was a homogeneous polysaccharide. Meanwhile, eight monosaccharides were identified in FAX by HPLC analysis, namely mannose, rhamnose, glucuronic acid, galacturonic acid, glucose, galactose, xylose, and arabinose (Figure 1b). The molar ratio of these monosaccharides was determined to be 0.2:0.2:0.2:0.3:0.3:1.0:3.1:2.3 (Table 1). Xylose and arabinose accounted for approximately 71% of the total monosaccharides, and the ratio of arabinose to xylose was calculated to be about 0.8, suggesting that FAX was an arabinoxylan with a relatively high degree of substitution [16]. Additionally, ferulic acid (FA) and minor amounts of diferulic acids (diFAs), including 8-5′, 5-5′, 8-*O*-4′, and 8-5′ benzofuran diFAs, were identified in the FAX sample (Figure 1c). The FA content was determined to be 33.3 µM, significantly higher than that of any detected diFAs (Table 2). The obtained FAX in this study possessed a ferulic acid composition and diferuloyl cross-linking architecture comparable to that of arabinoxylan previously prepared with GH30 1,4-β-xylanase [18]. Both FA and diFAs are important moieties in cereal arabinoxylans and play critical roles in arabinoxylan hydrogel formation via laccase-induced cross-linking [17,37]. Taken together, the enzymatically extracted FAX sample was a pure arabinoxylan with a high amount of FA moieties, which is beneficial for arabinoxylan hydrogel formation [18,19,20,28,38,39].

### 3.2. Dynamic Analysis of FAX Hydrogel Formation Under Optimal Conditions

The dynamic analysis of hydrogel formation was carried out in optimal conditions, i.e., the FAX concentration, enzyme concentration, pH, and reaction temperatures were set to 40 mg/mL, 10 mg/mL, pH 6, and 25 °C, respectively. The Q-ToF-MS chromatograms indicated that during the first 60 min of laccase-catalyzed gel formation, the amount of ferulic acid significantly decreased with a corresponding increase in several diFAs (Figure 2a). Specifically, the amount of FA decreased by approximately 11-fold (from 39 to 3.6 µM). In contrast, the concentrations of the 8-5′, 5-5′, 8-*O*-4′, and 8-5′ benzofuran diFA isomers increased by approximately 28.3, 5.0, 5.7, and 22.2-fold, respectively (Figure 2e). This transformation profile of FA into its diFA cross-links closely resembled what we previously reported for the laccase-induced gelation of FAX derived from a GH30 xylanase-extraction of corn bran [37,40]. It should be noted that the peak appearing at the retention time of about 7.5 min was not identified (Figure 2a). This peak was detected by MS at an *m*/*z* the same as the other four types of diFAs, and the fragment spectra were like the other diFAs. This evidence suggests that the unknown peak corresponds to an unidentified diFA isomer, the concentration of which also increased as the ferulic acid monomer was consumed. This hypothesis is supported by recent literature on wheat bran arabinoxylan, which highlights the strong involvement of other diFA isomers, such as 8-8′ and 5-5′, during the crosslinking process [19,32].

The gelation process of FAX was further investigated through time-resolved SEM to analyze the evolution of the FAX hydrogel network formation. In the early stage, i.e., 2–5 min, large pores and thin pore walls were observed on the surface and the large, irregular pores were also found in the cross-sectional structure, indicating incomplete cross-linking and weak hydrogel structures (Figure 2b,c). As the time increased to 10 and 20 min, pores on the surface of the FAX hydrogel became smaller and more evenly distributed, while the cross-sectional structure of the FAX hydrogel started to be more honeycomb-like with more defined pores and thickened walls (Figure 2b,c), revealing progressive network formation and enhanced cross-linking at this stage. At the final stage, i.e., 60 min, the FAX hydrogel surface was smooth with minimal pores, and the cross-sectional SEM showed a highly compact and uniform honeycomb-like structure (Figure 2b,c), confirming the establishment of a well-structured and stable hydrogel network. The changes in the microstructure of FAX hydrogel during dynamic hydrogel formation was in accordance with the data of the change of ferulic acid and diFAs described above (Figure 2a,e). Furthermore, the gelation kinetics further confirmed that the storage modulus (G′) values increased during the hydrogel formation from 0–60 min, and the loss modulus (G″) value reached a lag phase around 60 min (Figure 2d), indicating that sufficient time, more than 60 min, was required to form a hydrogel with satisfactory hydrogel appearance and stability (Figure 2f). A predominantly solid-like behavior was confirmed throughout the gelation process of 60 min, with the G′ values staying at least an order of magnitude higher than the G″ values (Figure 2d). The result of FAX hydrogel formation is in accordance with some previously reported studies [19,31,32,33,34]. Meanwhile, the final G′ value of the prepared FAX hydrogel in the present work was significantly higher than the G′ values reported for other laccase-catalyzed FAX hydrogels in the literature [19,31,32,33,34]. It is believed that the stable hydrogel network of the FAX hydrogel is related to a relatively high amount of FA moieties in the polysaccharide chains. Additionally, the FAX hydrogel also showed 10–100 folds higher G′ values compared to the xylan-based hydrogel prepared by other methods such as peroxidase/H_2_O_2_ system [32,41], diethylenetriaminepenta-acetic dianhydride (DTPA) [42] and co-polymerized hydrogel system with ethylene glycol diglycidyl ether (EGDE) as a crosslinker [43]. Taken together, the results may suggest the FAX hydrogel formed a mechanically robust structure (Figure 2d).

### 3.3. The Effect of Substrate Concentration on FAX Hydrogel Formation

#### 3.3.1. Rheological Properties of FAX Gel with Different Substrate Concentrations

To evaluate the effect of substrate concentration, i.e., FAX concentration on laccase-catalyzed FAX hydrogel formation, a series of substrate concentrations ranging from 10–50 mg/mL was investigated. Rheological properties, specifically the G′ and G″ values, were monitored over time to assess the gelation process. The gelation kinetics of FAX hydrogels was assayed via a dynamic rheological analysis (time sweep tests). The final G′ value of the indicated hydrogel strength increased significantly with concentration from 10 mg/mL (140.41 Pa) up to a peak at 30 mg/mL (1685.60 Pa) (Figure 3a). However, a further increase in concentration to 40 and 50 mg/mL did not enhance the final hydrogel strength significantly (Figure 3a). This result might be due to excessively dense hydrogel networks formed at higher concentrations, potentially leading to structural limitations and reduced network elasticity. The result also revealed that a higher FAX concentration (30–50 mg/mL) could accelerate the laccase-catalyzed hydrogel formation, which resulted in a shorter time when the hydrogels reached the lag phase (Figure 3a). The result indicated that the hydrogel formation of FAX hydrogels was greatly affected by the FAX concentration, and this may be explained by the fact that FAX at higher concentration exhibits more catalytic sites, i.e., feruloyl moieties for laccase enzyme. Previous studies also demonstrated that a higher density of arabinoxylan chains could increase the chances of the ferulic free radicals contacting each other to form covalent linkages [34,44]. After laccase-catalyzed gelation of FAX samples, these hydrogels were further characterized. As illustrated in Figure 3b, all tested FAX hydrogels exhibited significantly higher G′ values compared to the G″ values across the frequency range of 0.1–10 Hz, indicating that these hydrogels showed predominantly elastic properties typical of cross-linked hydrogel networks. As expected, the G′ values of FAX hydrogels increased with increasing substrate concentration. Notably, hydrogels formed at higher concentrations (40 and 50 mg/mL) exhibited substantially higher G′ values, indicating stronger elastic properties and stable hydrogel networks compared to those at lower concentrations (Figure 3b). The G′ values of FAX hydrogel at 50 mg/mL were always higher than 2100 Pa under the tested frequency ranging from 0.1–10 Hz, which were 4-fold and 10-fold higher than those of FAX hydrogel at 40 mg/mL and 10 mg/mL (Figure 3b). The strain sweep test was carried out to investigate the effect of FAX concentration on the mechanical stability and structural integrity of laccase-induced FAX hydrogels. The results demonstrated that the FAX hydrogels at lower concentrations (10–20 mg/mL) showed limited linear viscoelastic region (LVR) ranges, and the G′ values started to reduce at lower strain (Figure 3c), suggesting a weaker hydrogel network. As the concentration of FAX increased, particularly at concentrations above 30 mg/mL, a significantly wider LVR was observed (Figure 3c). The widest LVR was detected at the highest concentration tested (50 mg/mL), indicating remarkable structural stability and greater resistance to deformation under the stronger mechanical stress. Taken together, the hydrogel formation kinetics, elasticity, and structural stability of FAX hydrogels were affected by the substrate concentrations, and a high concentration (>30 mg/mL) facilitated the formation of a stable hydrogel.

#### 3.3.2. Water Holding Capacities of FAX Gel with Different Substrate Concentrations

As shown in Figure 3d, the WHC of FAX hydrogels exhibited a clear dependence on substrate concentration, increasing significantly as the FAX concentration increased from 10 to 50 mg/mL. The result may indicate that the FAX hydrogel at lower concentration exhibited a weaker and less compact network with limited ability to retain water, whereas the hydrogels at higher concentration showed a more stable hydrogel network with greater water retention. These results were in accordance with the rheological properties of these FAX hydrogels (Figure 3a–c).

#### 3.3.3. Microstructure of FAX Gel with Different Substrate Concentrations

The microstructure of FAX hydrogels was analyzed through a scanning electron microscopy (SEM) analysis. Both surface morphology and cross-sectional structures of these FAX hydrogels were analyzed, and the results revealed that the hydrogel network integrity and porosity were affected by the FAX concentrations (Figure 3e,f). Surface morphology images indicated that hydrogels formed at lower concentrations (10–20 mg/mL) possessed loose, porous structures with many irregular and large pores (Figure 3e). With the increase in FAX concentration (>30 mg/mL), progressively smoother and denser surfaces with fewer pores were observed (Figure 3e), suggesting a more robust hydrogel network. The cross-sectional structures of FAX hydrogels further confirmed the findings. The FAX hydrogels at lower concentrations (10–20 mg/mL) demonstrated loose, irregular, fibrous cross-sectional structures with large and unevenly distributed pores (Figure 3f), which indicated weaker hydrogel networks. In contrast, the FAX hydrogels at higher concentrations showed significantly more well-organized honeycomb-like structures with evenly distributed pores (Figure 3f). The microstructures of the FAX hydrogels at different concentrations were consistent with their rheological properties, as an increase in the concentration of FAX notably improved both the elasticity and stability of the hydrogels, suggesting a more densely crosslinked hydrogel network formed through laccase-mediated oxidative coupling (Figure 3a–c).

### 3.4. The Effect of Enzyme Concentration on FAX Hydrogel Formation

#### 3.4.1. Rheological Properties of FAX Gel with Different Enzyme Concentrations

The FAX concentration was set to 40 mg/mL, and the effect of laccase concentration (0.625–10 mg/mL) on hydrogel formation was investigated. Interestingly, the time sweep test revealed that the highest G′ (about 1030–1160 Pa) values were achieved in FAX hydrogels formed at lower laccase concentrations (0.625–1.25 mg/mL), while the FAX hydrogels at higher laccase concentration showed slightly lower G′ values at 60 min (Figure 4a). The gelation kinetics improved significantly with higher laccase concentrations, and the time when G′ values reached a log phase dramatically decreased from approximately 33.8 min at 0.625 mg/mL to below 10 min at 5 mg/mL and above (Figure 4a). This result indicated that higher laccase concentrations could facilitate hydrogel formation via enzymatic cross-linking. In the frequency sweep test, an increase in laccase concentration from 0.625 mg/mL to 5 mg/mL significantly enhanced the G′ values, suggesting improved elastic network formation (Figure 4b). However, no substantial differences in the G′ values (1800–1900 Pa at max) were observed between hydrogels formed at 5 mg/mL and 10 mg/mL laccase, indicating that a further increase in enzyme concentration did not significantly enhance the elastic properties of the FAX hydrogels. At the strain sweep test, FAX hydrogels at all enzyme concentrations showed a relatively wide LVR (Figure 4c). Interestingly, the LVR range decreased progressively with increasing laccase concentrations. Specifically, hydrogels formed at lower enzyme concentrations (0.625–1.25 mg/mL) exhibited a relatively broader LVR (Figure 4c), which may indicate a more stable hydrogel network and higher resistance to deformation under high strain. In contrast, higher laccase concentrations (≥5 mg/mL) resulted in significantly narrower LVRs, suggesting that rapid enzymatic cross-linking appeared to increase the network rigidity and reduced the deformability.

#### 3.4.2. Water Holding Capacities of FAX Gel with Different Enzyme Concentrations

The WHC of FAX hydrogels with different laccase concentrations is shown in Figure 4d. The results demonstrated that FAX hydrogels with lower laccase concentrations (0.625–1.25 mg/mL) exhibited higher WHC (∼80%), while an increase in enzyme concentration significantly reduced the WHC to 68% (Figure 4d). The high laccase concentration could catalyze the feruloyl cross-linking more rapidly; however, it was also likely to form a more rigid hydrogel network with reduced water retention capacity.

#### 3.4.3. Microstructure of FAX Gel with Different Enzyme Concentrations

The microstructure of FAX hydrogels analyzed by SEM revealed that the significant differences in hydrogel surface morphology and cross-sectional structure correlated with enzyme concentration. Surface morphology images demonstrated that hydrogels formed at lower enzyme concentrations, i.e., 0.625 mg/mL displayed a porous and loosely-structured surface, indicating weaker network integrity (Figure 4e). With increasing laccase concentration (≥1.25 mg/mL), a decrease in pores was observed, which resulted in a denser, smoother, and more continuous hydrogel surface structure (Figure 4e). At higher enzyme concentrations (5 and 10 mg/mL), the hydrogels exhibited highly compact network structures with almost no pores (Figure 4e). These overly dense and rigid structures may explain the limited water penetration and retention (Figure 4d). Cross-sectional SEM images further confirmed these findings, indicating that lower enzyme concentrations produced hydrogels with large, irregular pores and loosely-organized structures (Figure 4f). With an increase in enzyme concentration (up to 10 mg/mL), the FAX hydrogels showed a cross-sectional structure with a well-organized honeycomb-like morphology, indicating enhanced rigidity and structural density, corresponding to the previously observed rheological properties (Figure 4a–c). In summary, an increase in the laccase concentration could improve the gel properties and microstructure at lower concentration range (from 0.625–1.25 mg/mL); however, when the laccase concentration exceeded 1.25 mg/mL, this effect was substantially weakened. These results demonstrate that a laccase concentration of 1.25 mg/mL is sufficient to obtain satisfactory hydrogels, which has significant implications for large-scale production in food industries under consideration of lower cost.

### 3.5. The Effect of Reaction pH on Arabinoxylan Hydrogel Formation

#### 3.5.1. Rheological Properties of FAX Gel with Different Reaction pH

The FAX concentration and laccase concentration was set to 40 mg/mL and 10 mg/mL, respectively, and the effect of reaction pH values (3–8) on hydrogel formation was investigated. The FAX hydrogel formed under an acidic condition, i.e., pH = 3–4 showed relatively slower gelation kinetics compared to the hydrogels formed under a pH of 5–8, which spent more than 60 min to reach the lag phase, indicating slower oxidative cross-linking (Figure 5a). Interestingly, the final G′ values of FAX hydrogels at low pH values (3–4) were slightly higher than those of FAX hydrogels at pH values of 5 and above (Figure 5a). The hydrogel formation was significantly facilitated with the increase in pH from 3–6 (Figure 5a), and then decreased as the pH increased to 8, suggesting the optimal pH range of the laccase studied in the present work was about pH 5–6. Additionally, the frequency sweep test revealed that hydrogels formed at pH 5–6 exhibited significantly higher G′ values (around 1350–1500 Pa) compared to the FAX hydrogels formed at other pH values (Figure 5b), which may indicate the formation of robust and well-developed hydrogel networks. A further increase to pH 8 decreased the G′ values to approximately 420 Pa. Meanwhile, the FAX hydrogels at pH 3–4 showed lower G′ values of approximately 240–260 Pa (Figure 5b). The strain sweep test demonstrated that FAX hydrogels at pH 5–6 exhibited significantly higher G′ values across all strain ranges, compared to other hydrogels (Figure 5c). Meanwhile FAX hydrogels formed at pH 3–6 showed relatively broad LVRs, while the hydrogels at pH 7–8 were slightly narrower (Figure 5c). This result is in good agreement with a previous report, which showed that when using the same laccase from *Trametes versicolor* with a wheat bran arabinoxylan substrate, the resulting hydrogel exhibited significantly superior rheological properties at pH 5 compared to those formed at pH 2 and pH 7 [19].

#### 3.5.2. Water Holding Capacities of FAX Gel with Different Reaction pH

The WHC of FAX hydrogels at pH 5–7 (around 77–80%) were slightly higher compared to other FAX hydrogels (Figure 5d). The results indicated that the WHC appeared not to be significantly affected by the reaction pH, despite the differences in rheological properties among all FAX hydrogels (Figure 5b,c).

#### 3.5.3. Microstructure of FAX Gel with Different Reaction pH

The SEM results revealed a clear difference in surface morphology and cross-sectional structure among the FAX hydrogels at different pH values. At low pH (pH 3–4), hydrogels exhibited a relatively smooth and compact surface with few visible pores. At pH 6, the FAX hydrogel showed a very smooth surface with few visible pores (Figure 5e), confirming the formation of a highly cross-linked and stable hydrogel network. This was in accordance with the highest G′ and moderate LVR (Figure 5b,c), which may indicate a good hydrogel strength and flexibility. As the pH increased to 8, the surfaces of the FAX hydrogels exhibited increased porosity (Figure 5e), suggesting reduced cross-linking efficiency and weakened hydrogel network integrity. Meanwhile, FAX hydrogels at pH 3–5 showed cross-sectional structures with large, irregularly-shaped pores and unevenly-distributed networks (Figure 5e), indicating an insufficiently cross-linked structure. The FAX hydrogel at pH 6 exhibited a highly organized honeycomb-like structure with small and evenly distributed pores, corresponding to the highest G′ and great mechanical properties (Figure 5b,c). In addition, irregular and partially collapsed pores were observed in FAX hydrogel at pH 8, indicating a less stable network structure. These findings suggested that the laccase used prefers a mild pH around 6–7; either alkaline or acidic conditions may reduce the formation of a stable hydrogel network of FAX.

#### 3.5.4. Hydrogel Formation Under Alkaline Conditions

As the reaction pH of laccase-catalyzed hydrogel formation of FAX increased to 9 and 10, we found that hydrogel formation was not able to succeed, and the FAX hydrogel solutions remained in liquid phase (Figure 5h). As shown in the time sweep test result, both G′ and G″ values in FAX hydrogels at pH of 9 and 10 were significantly lower than other FAX hydrogels, and the similar G′ and G″ values further indicated these FAX samples were in liquid-like solutions (Figure 5a). One hypothesis of this phenomenon may be the liberation of feruloyl moieties from arabinoxylan chains, causing a lack of ferulic acid for laccase-catalyzed cross-linking. Hence, the bound ferulic acids and diferulic acids were determined via Q-ToF-MS analysis, and the results indicated a high amount of ferulic acid from FAX after an incubation with laccase at pH 9 and 10 for more than 70 min (Figure 5g). In addition, no detectable ferulic acid or diFAs were observed in FAX hydrogel solutions at pH 9 and 10 (Figure 5g). These results suggested that the feruloyl moieties of FAX were not liberated from the polysaccharide chains under the alkaline conditions, which may suggest the unsuccessful hydrogel formation at pH 9 and 10 may be explained by an inactivation of laccase enzyme under the alkaline conditions. Taking the results together, the pH could significantly affect the FAX hydrogel formations, and a mild pH condition may help the formation of a stable hydrogel. It should also be noted that the addition of pH stabilizer may improve the enzyme activity of laccase under rough pH conditions.

### 3.6. The Effect of Reaction Temperature on Arabinoxylan Hydrogel Formation

#### 3.6.1. Rheological Properties of FAX Gel with Different Reaction Temperature

The FAX concentration, enzyme concentration, and pH were set to 40 mg/mL, 10mg/mL, and pH 6, respectively, and the effect of reaction temperature (15–55 °C) on FAX hydrogel formation was evaluated. The gelation kinetics of FAX hydrogel formations were significantly affected by the reaction temperatures during the time sweep test (0–70 min) (Figure 6a). An increase in temperature appeared to facilitate the FAX hydrogel formation, and hydrogels at 15 °C and 25 °C reached the lag phase over a longer time. Interestingly, the hydrogels at high reaction temperatures, i.e., 45 °C and 55 °C, showed a second sharp increase in the curve when their G′ values reached a lag phase (Figure 6a), and this may be due to the effect of water evaporation in the FAX hydrogel solutions at 45 °C and 55 °C during the time sweep test. Meanwhile, the frequency sweep test revealed that FAX hydrogels at 15 °C and 25 °C exhibited significantly higher G′ values compared to FAX hydrogels at higher temperatures (up to 55 °C) (Figure 6b), suggesting a more stable hydrogel network. Furthermore, higher G′ values were also observed in FAX hydrogels at 15 °C and 25 °C under the strain sweep test (Figure 6c). Additionally, broader LVRs were found in FAX hydrogels at lower temperatures (<35 °C) (Figure 6c), indicating the hydrogels formed at low temperature may have a more stable hydrogel network and exhibit a greater resistance to deformation under high stress.

#### 3.6.2. Water Holding Capacities of FAX Gel with Different Reaction Temperature

As shown in Figure 6d, the FAX hydrogels formed at lower temperatures (15–25 °C) had higher WHC, suggesting greater water retention capacity.

#### 3.6.3. Microstructure of FAX Gel with Different Reaction Temperature

The SEM results revealed a clear difference in surface morphology and cross-sectional structure among FAX hydrogels formed at different temperatures. The results revealed that the hydrogel surface of FAX hydrogel at 15 °C was rough, with some visible folds and irregular pore distribution (Figure 6e), which indicated relatively weak cross-linking and poor network integrity. The hydrogel at 25 °C showed a very uniform and smooth surface with minimal visible pores (Figure 6e), suggesting that this temperature could produce the best cross-linked and homogeneous hydrogel network. A further increase in reaction temperature caused a less stable hydrogel network, while the increased porosity and partial collapse in FAX hydrogel at 55 °C may suggest potential network deformation due to excessive temperature (Figure 6e). The cross-sectional SEM analysis further indicated that only the FAX hydrogel at 25 °C exhibited a well-organized honeycomb-like network with uniform pores (Figure 6f). The FAX hydrogel at 15 °C and 35 °C showed honeycomb-like networks; however, the pores were larger and unevenly distributed (Figure 6f), indicating a less stable cross-sectional structure. Partial structural collapse and irregular pore distribution were observed in FAX hydrogels at 45 °C and 55 °C, indicating a decrease in hydrogel stability (Figure 6f). Taking the results together, the most satisfactory reaction temperature was observed as 25 °C. A high reaction temperature (>35 °C) could cause a worse FAX hydrogel formation with a less stable network structure, and this may be explained by water evaporation at high temperatures. However, a relatively low reaction temperature, e.g., 15 °C, may decrease the enzyme activity and result in worse hydrogel formation. The results are in agreement with other reported studies that hydrogels are more likely to form at lower temperatures [45].

### 3.7. Characteristics of the Hydrogel Structure Under Optimal Experimental Conditions

The FT-IR spectra of FAX and FAX hydrogels are shown in Figure 7a. Typical absorption bands of FAX appeared at 1200–800 cm^−1^ with the main band at 1034 cm^−1^, which might be related to C-OH bending [46,47]. The absorption peak at around 1250 cm^−1^ is the C-O stretching of ether bonds in the cross-linked structure of xylan or ferulic acid [48], and the absorption peak at around 1631 cm^−1^ is the amide I, which verifies the presence of protein and agrees with the results of the protein content of 2.3% ± 0.4 in the previous analysis. The absorption peak of the carboxyl group was around 1730 cm^−1^ but the peak was absent in the FAX hydrogel. The broad absorption peak at 3425 cm^−1^ showed the stretching of -OH groups [49]. It was also found that the FAX FTIR spectra of AX treated with and without laccase were similar and showed similar peaks at wavelengths of 1034 cm^−1^, 1631 cm^−1^, 2936 cm^−1^, and 3425 cm^−1^ [50].

Most arabinoxylans in nature have an amorphous structure due to the fact that the branched chains prevent the arabinoxylan main chains from approaching each other and forming crystalline zones. The XRD results (Figure 7b) showed two diffraction peaks at 2θ = 10° and 2θ = 20° for the FAX hydrogel, but since both peaks were broad, it indicated that the FAX hydrogel was mainly amorphous [51]. The amorphous form of the FAX form provided better bioavailability.

Nuclear magnetic resonance (NMR) relaxometry is used as a non-destructive method to probe the mobility and distribution of water molecules in gel structures. The spin–spin relaxation time (T2) is a commonly used parameter for analyzing hydrogen proton relaxation and interpreting the interaction between water and biopolymers. In general, a shorter T2 indicates a stronger interaction between water molecules and the gel matrix. Figure 7c shows the T2 variations of the FAX hydrogel samples. Three independent water peaks—T2b (0.1–1 ms), T21 (1–100 ms), and T22 (100–1000 ms)—were observed, which referred to strongly bound water, weakly bound water, and free water, respectively. T2b accounted for 0.486% ± 0.056%, T21 accounted for 1.512% ± 0.224%, and T22 accounted for 98.001% ± 0.168%, which suggests that only small amounts of strongly bound and weakly bound water were contained in the FAX hydrogels. Results also indicated that most of the water was distributed in the gel network structure in the form of free water, which is in accordance with a previous study [24].

## 4. Conclusions

In the present study, we systematically investigated the laccase-catalyzed gelation of soluble feruloylated arabinoxylan (FAX) enzymatically extracted from corn bran. The gelation mechanism was confirmed to be oxidative coupling of ferulic acid monomers into diferulic acid cross-links, i.e., 8-5′, 5-5′, 8-*O*-4′, and 8-5′ benzofuran diferulic acids. The systematic investigation established that processing parameters, i.e., FAX concentration, laccase concentration, reaction pH, and reaction temperature, critically govern the final hydrogel characteristics. SEM analysis revealed a denser and more uniform honeycomb-like microstructure of the FAX hydrogel, corresponding to a mechanically robust hydrogel, as being achieved under optimal conditions: 30 mg/mL FAX concentration, 1.25 mg/mL laccase, pH 6, and 25 °C. Deviations from these optimal conditions led to distinct structural changes; for instance, lower substrate concentrations resulted in loose, irregular porous structures, while higher laccase concentrations, produced overly dense and rigid networks. Similarly, non-optimal pH, e.g., pH 9 and 10, and unsatisfactory temperatures led to less organized or partially collapsed microstructures. The prepared FAX hydrogels existed mainly in an amorphous state and the water in them was mainly in the form of free water. In conclusion, this work provided a vital theoretical basis for the rational design and controllable fabrication of FAX hydrogels with tailored microstructures and functionalities, which may be beneficial for the food industries.

## Figures and Tables

**Figure 1 foods-14-02819-f001:**
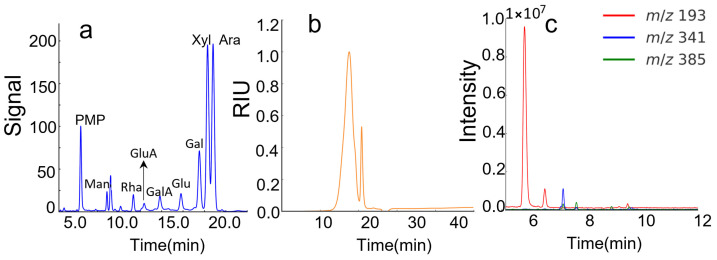
(**a**) SEC chromatogram of FAX; (**b**) monosaccharide compositions of FAX; (**c**) chromatographic profiles of ferulic acid and four types of diFAs, i.e., 8-5′, 5-5′, 8-*O*-4′, and 8-5′ benzofuran diFA of FAX. Color coding is used in all chromatograms: FA, red, *m*/*z* 193; FA, 8-5′ diFA (green, *m*/*z* 385); 5-5′ diFA (green, *m*/*z* 385); 8-*O*-4′ diFA (green, *m*/*z* 385) and 8-5′ benzofuran diFA (blue, *m*/*z* 341). The peak appearing at retention time about 7.5 min is unknown but has an *m*/*z* of 385 and similar fragments as other diFAs.

**Figure 2 foods-14-02819-f002:**
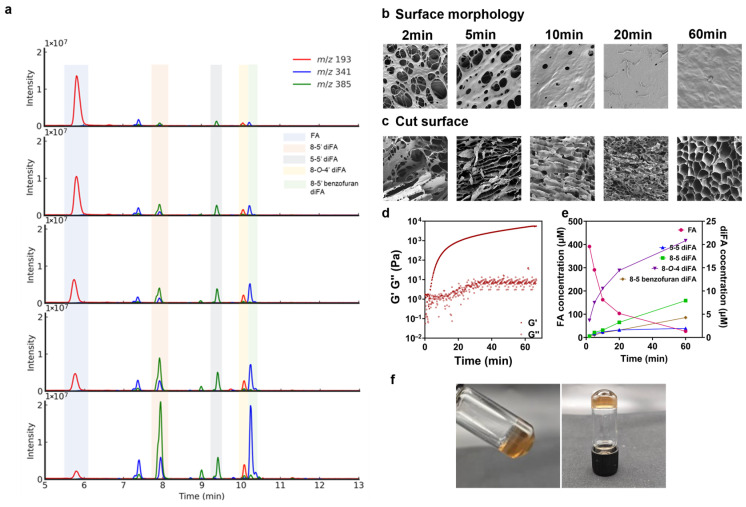
(**a**) Dynamic change of ferulic acid and four types of diFAs during the FAX hydrogel formation (2, 5, 10, 20, and 60 min from top to bottom); (**b**,**c**) SEM images showing surface morphology and cross-sectional structures of FAX hydrogels collected at different time points during the hydrogel formation; (**d**) gelation kinetics of hydrogel formation of FAX hydrogels under a time sweep test; (**e**) changes of amount of ferulic acid and diFAs during FAX hydrogel formation; (**f**) appearance of FAX hydrogel.

**Figure 3 foods-14-02819-f003:**
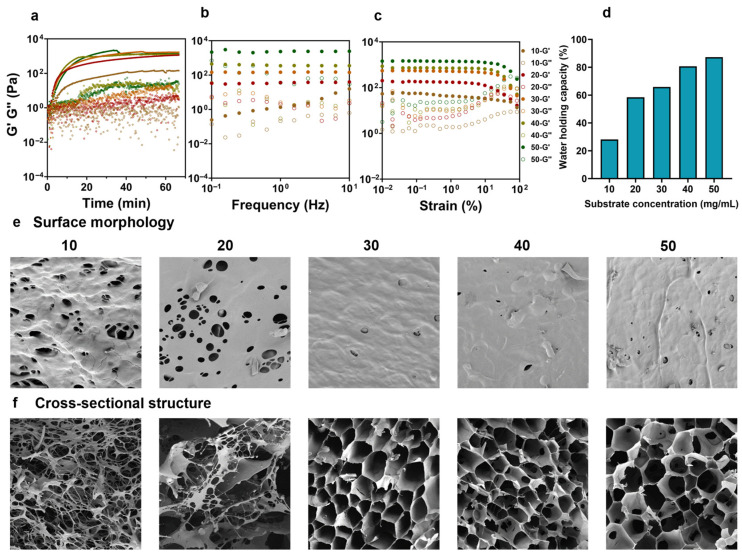
(**a**) Gelation kinetics of hydrogel formation of FAX at different substrate concentrations (10–50 mg/mL) under a time sweep test (0–70 min); (**b**) frequency sweep test of FAX hydrogels at different substrate concentrations; (**c**) strain sweep test of FAX hydrogels at different substrate concentrations; (**d**) water holding capacities (WHC) of FAX hydrogels at different substrate concentrations; (**e**,**f**) SEM images showing surface morphology and cross-sectional structures of FAX hydrogels at different substrate concentrations.

**Figure 4 foods-14-02819-f004:**
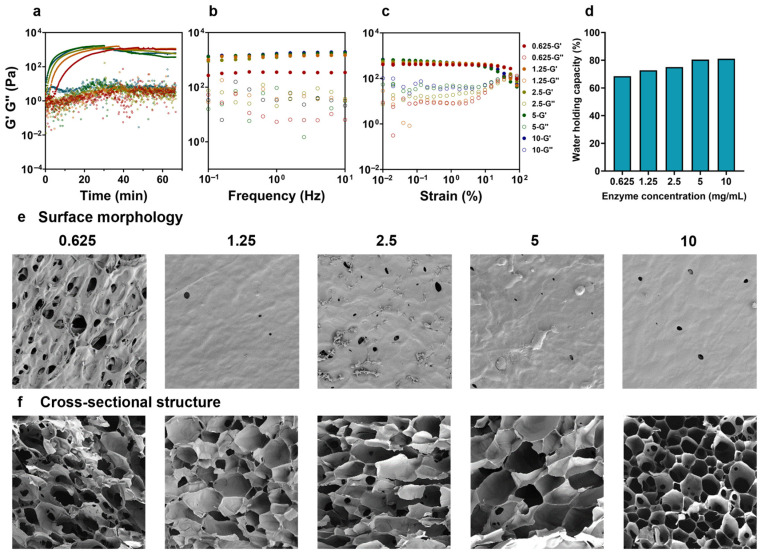
(**a**) Gelation kinetics of hydrogel formation of FAX with laccase at different enzyme concentrations (0.625–10 mg/mL) under a time sweep test (0–70 min); (**b**) frequency sweep test of FAX hydrogels with laccase at different enzyme concentrations; (**c**) strain sweep test of FAX hydrogels with laccase at different enzyme concentrations; (**d**) water holding capacities (WHC) of FAX hydrogels with laccase at different enzyme concentrations; (**e**,**f**) SEM images showing surface morphology and cross-sectional structures of FAX hydrogels with laccase at different enzyme concentrations.

**Figure 5 foods-14-02819-f005:**
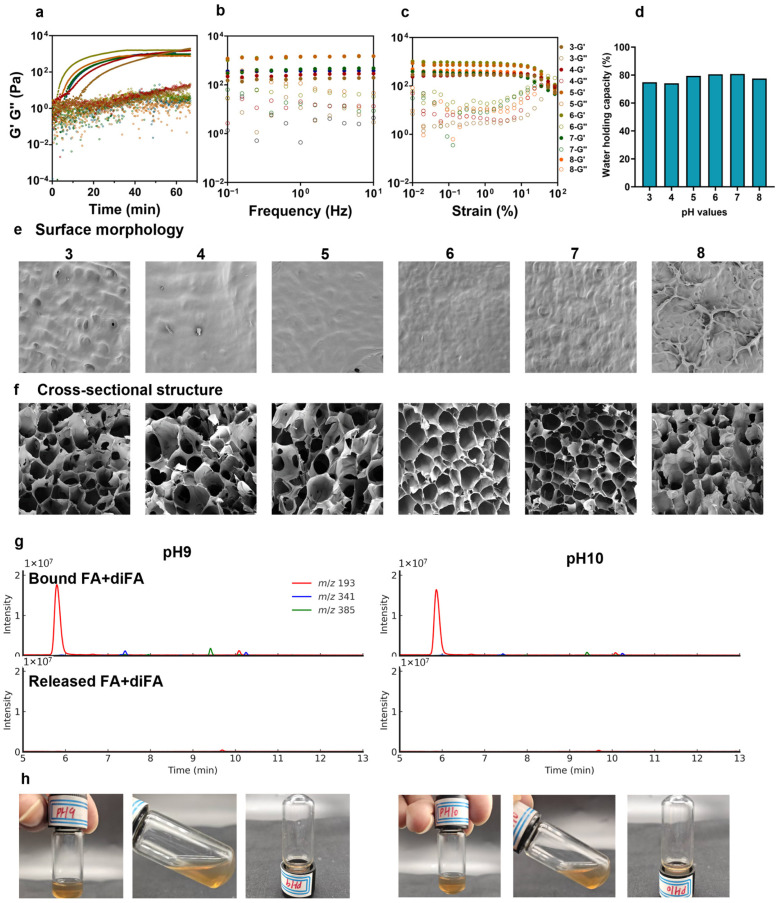
(**a**) Gelation kinetics of hydrogel formation of FAX under different reaction pH (3–8) under a time sweep test (0–70 min); (**b**) frequency sweep test of FAX hydrogels under different reaction pH; (**c**) strain sweep test of FAX hydrogels under different reaction pH; (**d**) water holding capacities (WHC) of FAX hydrogels under different reaction pH; (**e**,**f**) SEM images showing surface morphology and cross-sectional structures of FAX hydrogels under different reaction pH; (**g**) Q-ToF-MS chromatograms of bound ferulic acid and diFAs and released ferulic acid and diFAs in FAX hydrogel solutions under reaction pH 9 and 10; (**h**) appearance of FAX hydrogel solutions under reaction pH 9 and 10.

**Figure 6 foods-14-02819-f006:**
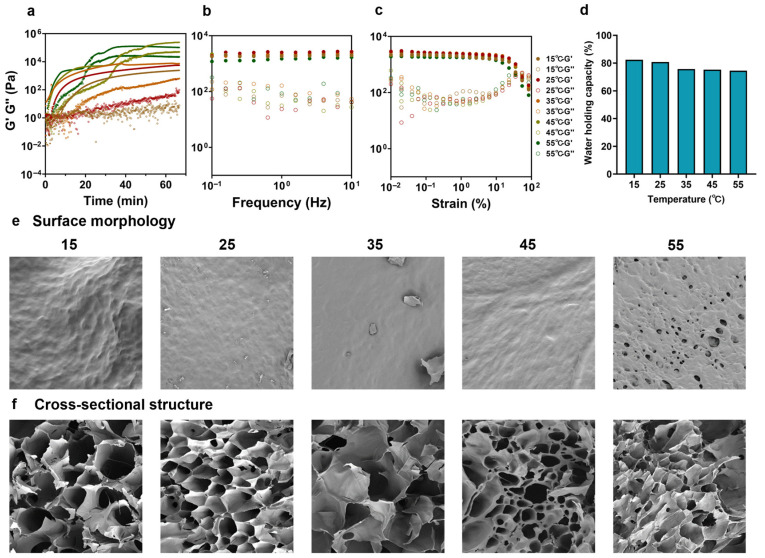
(**a**) Gelation kinetics of hydrogel formation of FAX hydrogels under different temperatures (15–55 °C) under a time sweep test (0–70 min); (**b**) frequency sweep test of FAX hydrogels under different temperatures; (**c**) strain sweep test of FAX hydrogels under different temperatures; (**d**) water holding capacities (WHC) of FAX hydrogels under different temperatures; (**e**,**f**) SEM images showing surface morphology and cross-sectional structures of FAX hydrogels under different temperatures.

**Figure 7 foods-14-02819-f007:**
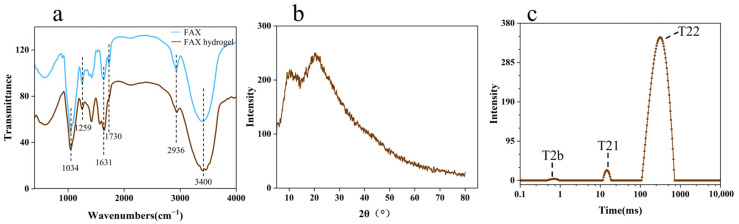
(**a**) Infrared spectra of the optimal conditioned FAX hydrogel and FAX; (**b**) XRD plots of optimal conditioned FAX hydrogels; (**c**) low-field NMR image of optimal conditioned FAX hydrogel.

**Table 1 foods-14-02819-t001:** Extraction yield, total polysaccharide content, total protein content molecular weight, molar ratio of monosaccharide compositions of FAX.

	Extraction Yield	Total Polysaccharide	Total Protein	*M*_w_ (kDa)	Molar Ratio of Monosaccharide Compositions							
					Man	Rha	GlcA	GalA	Glc	Gal	Xyl	Ara
FAX	0.75% ± 0.1	80.1% ± 1.8	2.3% ± 0.4	339.8 ± 12.1	0.2	0.2	0.2	0.3	0.3	1.0	3.1	2.3

**Table 2 foods-14-02819-t002:** Ferulic acid and four types of diferulic acids in FAX and FAX hydrogels.

	FA (µM)	8-5′ diFA (10^−3^ µM)	5-5′ diFA (10^−3^ µM)	8-*O*-4′ diFA (10^−3^ µM)	8-5′ Benzofuran diFA (10^−3^ µM)
FAX	33.3	14.4	6.5	208.2	3.6

Note: The FA and diFA contents were calculated as in 1 mg/mL FAX.

## Data Availability

The original contributions presented in this study are included in the article. Further inquiries can be directed to the corresponding authors.

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
