# Peer review of "Hydrogel Formation of Enzymatically Solubilized Corn Bran Feruloylated Arabinoxylan by Laccase-Catalyzed Cross-Linking"

_foods, 2025, doi:10.3390/foods14162819_

Round 1

Reviewer 1 Report

Comments and Suggestions for Authors

The paper investigates the laccase-catalyzed hydrogel formation of feruloylated arabinoxylan (FAX) derived from corn bran. Emphasizing the relationship between processing parameters and hydrogel properties, the authors utilize systematic experimentation to elucidate the gelation mechanism, the impact of various factors (such as concentration, enzyme dosage, pH, and temperature), and the resultant microstructural characteristics of the hydrogels produced.

 The authors clearly elucidate the mechanism of hydrogel formation as oxidative coupling of ferulic acid into diferulic acids, enhancing the academic understanding of this process.

While the paper discusses the potential implications for the food industry, it would benefit from examples of specific applications or proposed experiments to test these applications further.

The paper primarily focuses on laccase-mediated hydrogel formation without comparing it to other enzymatic or non-enzymatic methods for producing hydrogels. Including such comparisons could provide a broader perspective on the advantages of the presented approach.

It is recommend to use similar works such as rice bran gel as “Rheology and microstructure of binary mixed gel of rice bran protein–whey: effect of heating rate and whey addition” which will be helpful in the results and comparing with this work.

M&M: Section 2.6.: what’s the gap size? Did you use normal PP or serrated PP?

The resolution of the figures is low, especially for the graphs and it is difficult to follows.

The findings indicate a trade-off regarding enzyme concentration and gel properties. While this is addressed, a deeper analysis of how this might influence industrial scalability (e.g., cost considerations) would enrich the discussion.

Although SEM is employed effectively, additional characterization techniques (e.g., atomic force microscopy, differential scanning calorimetry) could provide further insights into structural stability and thermal properties.

The authors can also benefits from using the work “Effect of inulin/kefiran mixture on the rheological and structural properties of mozzarella cheese” in this paper.

The study identifies an optimal pH range but notes challenges with higher pH values. A more in-depth exploration of enzyme behavior and other potential stabilizers at varying pH levels could yield additional insights.

Overall, the paper presents a valuable contribution to the field of hydrogel research, particularly concerning polysaccharide-based materials in the food industry. It combines rigorous experimental analysis with practical applications, providing a robust foundation for future studies. Addressing the highlighted weaknesses will further enhance the paper’s impact and applicability in both academic and industrial contexts.

Reviewer 2 Report

Comments and Suggestions for Authors

In this work the authors reported that "hydrogel formation of enzymatically solubilized corn bran feruloylated arabinoxylan by laccase-catalyzed cross-linking." Overall, this study was seriously done with a significant amount of experimental evidence. However, the following comments mentioned below should be addressed before recommendation for publication.

Comments

  1. How was the DCB converted to feruloylated arabinoxylan? Is it due to the addition of TlXyn11? Please make it clear. Furthermore, how was the TlXyn11 concentration fixed?
  2. What was the gelation time with different optimization conditions?
  3. The functional and chemcial properties of hydrogel are missing. For example, XRD, FTIR...
  4. How could these hydrogels be useful in food industries?

Round 2

Reviewer 2 Report

Comments and Suggestions for Authors

The authors seriously revised their manuscript based on the reviewer's comment.